# Microfluidics Biocatalysis System Applied for the Synthesis of N-Substituted Benzimidazole Derivatives by Aza-Michael Addition

Rong-Kuan Jiang [1,†], Yue Pan [1,†], Li-Hua Du [1,*], Ling-Yan Zheng [1], Zhi-Kai Sheng [1], Shi-Yi Zhang [1], Hang Lin [1], Ao-Ying Zhang [1], Han-Jia Xie [1], Zhi-Kai Yang [1] and Xi-Ping Luo [2,*]

1   College of Pharmaceutical Science, Zhejiang University of Technology, Hangzhou 310014, China
2   Zhejiang Provincial Key Laboratory of Chemical Utilization of Forestry Biomass, Zhejiang A&F University, Hangzhou 311300, China
*   Correspondence: orgdlh@zjut.edu.cn (L.-H.D.); luoxiping@zafu.edu.cn (X.-P.L.);
    Tel.: +86-189-690-693-99 (L.-H.D.)
†   These authors contributed equally to this work and should be considered co-first authors.

**Abstract:** Benzimidazole scaffolds became an attractive subject due to their broad spectrum of pharmacological activities. In this work, a methodology was developed for the synthesis of N-substituted benzimidazole derivatives from benzimidazoles and $\alpha$, $\beta$-unsaturated compounds (acrylonitriles, acrylate esters, phenyl vinyl sulfone) catalyzed by lipase TL IM from *Thermomyces lanuginosus* in continuous-flow microreactors. Investigations were conducted on reaction parameters such as solvent, substrate ratio, reaction temperature, reactant donor/acceptor structures, and reaction time. The transformation is promoted by inexpensive and readily available lipase in methanol at 45 °C for 35 min. A wide range of $\beta$-amino sulfone, $\beta$-amino nitrile, and $\beta$-amino carbonyl compounds were efficiently and selectively synthesized in high yields (76–97%). All in all, a microfluidic biocatalysis system was applied to the synthesis of N-substituted benzimidazole derivatives, and could serve as a promising fast synthesis strategy for further research to develop novel and highly potent active drugs.

**Keywords:** enzymatic synthesis; benzimidazole derivatives; continuous-flow reaction technology; continuous-flow microreactor; aza-Michael addition

## 1. Introduction

Heterocyclic compounds are well-documented pharmaceutically active substances that have been discovered to be crucial in drug design and development. Benzimidazole, as an aromatic heterocyclic organic compound, has received extensive attention in the field of drug research and development [1], due to its wide range of biological activities. Benzimidazole compounds with various substituents have been proven to possess obvious anti-inflammatory pain [2], anti-viral [3], anti-tumor [4], anti-hypertension [5], anti-diabetic [6], anti-HIV [7,8], excellent anti-fungal and bacterial activity [9–14], and other effects. Many drugs containing a benzimidazole structure have been used in clinical therapy, such as triclobendazole (anti-helminth drug and anti-fungal infection) [15], candesartan (anti-hypertension) [16], omeprazole (proton pump inhibitor) [17], bendamustine (anti-cancer) [18], selumetinib (first FDA-approved drug for neurofibromatosis type (1) [19], astemizole (histamine receptor antagonist) [20], and albendazole (insecticide) [21] (Figure 1).

The construction of C-N bonds is the key to synthesizing N-substituted benzimidazole derivatives. Among the methods that have been developed, the aza-Michael addition reaction of nitrogen nucleophiles to $\alpha$, $\beta$-unsaturated compounds has been demonstrated to be one of the simplest and most effective methodologies for the corresponding adducts, such as $\beta$-amino acids, $\beta$-amino nitrile, $\beta$-amino ester, and $\beta$-amino sulfone [22–24]. Traditionally, aza-Michael reactions proceed under strongly acidic or basic conditions that exhibited

poor compatibility with various substrate functional groups and generated side products, e.g., polymeric compounds; sometimes, a high temperature and long reaction time are required [25]. Furthermore, these methods are limited in substrate scope to aliphatic amine additions. In order to develop a more efficient aza-Micheal addition reaction suitable for wide substrate scope, a variety of metal catalysts, including $Cs_2CO_3$, $Cu(OAc)_2$, Pd, $Pd(OAc)_2$ [26], $Pd_2(dba)_3$ [27], $AgO_3SCF_3$ [28], and $AgSbF_6$ [29], were intensively developed and used to catalyze the conjugate addition of benzimidazoles to electron-deficient compounds. Despite great advances in metal-catalyzed aza-Michael additions, the use of expensive precious metals and sometimes the air- and moisture-sensitive metal catalysts raises the expense and difficulty of the reaction. In addition, these metal–catalyst methods are not suitable for the addition of less nucleophilic aryl amines or benzimidazoles.

**Figure 1.** Drugs possessing benzimidazole motifs.

Enzymes, as excellent biocatalysts, can be applied to many various reactions, including kinetic resolution, esterification, hydrolysis, aminolysis, and transesterification [30–34]. The use of enzymes as catalysts for the preparation of novel compounds has attracted much attention in the fields of organic synthesis, pharmaceuticals, petrochemicals, and materials over the past few years [30–33]. Among the numerous protocols involving aza-Michael additions, enzyme-catalyzed processes are particularly attractive due to their high efficiency and specificity. The first example of Michael addition catalyzed by enzymes was reported in 1988 by Kitazume et al., (E)-ethyl 3-(trifluoromethyl)- and 2-(trifluoromethyl)-propenates are readily converted to chiral Michael adducts via the addition of thiols or dialkylamines with the presence of lipase PL 266 and lipase PL 679 (from *Alcaligenes sp*) and lipase AL 865 (from *Achromobacter*) [35]. Novozyme 435 or Lipozyme RM IM was used to catalyze Michael addition of various primary and secondary amines to acrylonitrile, but the side product of aminolysis occurred when amines were added to the substituted acrylates. Then, Michael addition reactions of substituted acrylates such as methyl crotonate and methyl methacrylate catalyzed by lipases from *Pseudomonas stutzeri* (PSL) and *Chromobacterium viscosum* (CvL) were reported [36]. However, these enzymatic Michael addition reactions have a common point of requiring a long reaction time to obtain the desired yields. In previous studies, we reported lipase TL IM-catalyzed Michael addition reactions of pyrimidines, anilines, and the enzymatic Michael addition of imidazoles and acrylates in DMSO solvent [37–39]. Therefore, in this study, we decided to study further the aza-Michael addition reactions of benzimidazoles catalyzed by lipase to select greener solvents, simplify the steps of product purification, and synthesize a series of N-substituted benzimidazole derivatives.

Continuous-flow reactors have significant processing advantages, including improved thermal management, mixing control, and the application of extreme reaction conditions compared to stirred tank reactors, which can greatly intensify the synthetic processes and increase the reaction efficiency [40–44]. In 2007, the Green Chemistry Institute (GCI), part

of the American Chemical Society (ACS), set up a roundtable in conjunction with several pharmaceutical corporations. The roundtable listed several key areas where research was required to facilitate the development of sustainable manufacturing, and the importance of continuous processing was acknowledged [45]. With these factors in mind, there has been renewed interest in the development of sustainable processes, with many of 'big pharma' looking towards new techniques for both research and production [46]. Many drugs containing benzimidazole motifs have been used in clinical therapy. It is worthwhile to develop a greener, more effective technology for the construction of benzimidazole derivatives with potential pharmaceutical activities through continuous enzymatic processing.

The high selectivity of enzymes combined with the high efficiency of continuous-flow technology may reinforce transformations of benzimidazole derivatives. Many works on drug intermediates synthesis catalyzed by enzymes using continuous-flow technology were developed [37–39,47,48]. Given that benzimidazoles are attractive scaffolds with a range of pharmacological activities, it is imperative to find a more convenient and sustainable manufacturing technology to build the compounds library under the circumstances. In this work, we report a microfluidic biocatalysis system applied for the synthesis of N-substituted benzimidazole derivatives by aza-Michael addition. Benzimidazoles (benzimidazole, 2-chlorobenzimidazole, 2-methylbenzimidazole), as less nucleophilic agents, reacted with several α, β-unsaturated compounds (2-chloroacrylonitrile, acrylonitrile, methyl acrylate, methyl methacrylate, phenyl vinyl sulfone) catalyzed by lipase TL IM from *Thermomyces lanuginosus* in continuous-flow microreactors were studied (Scheme 1). Reaction parameters including solvent, substrate ratio, reaction temperature, reactant donor/acceptor structures, and reaction time were investigated. A series of different N-substituted benzimidazole derivatives were synthesized efficiently with high yields in this continuous-flow enzymatic strategy.

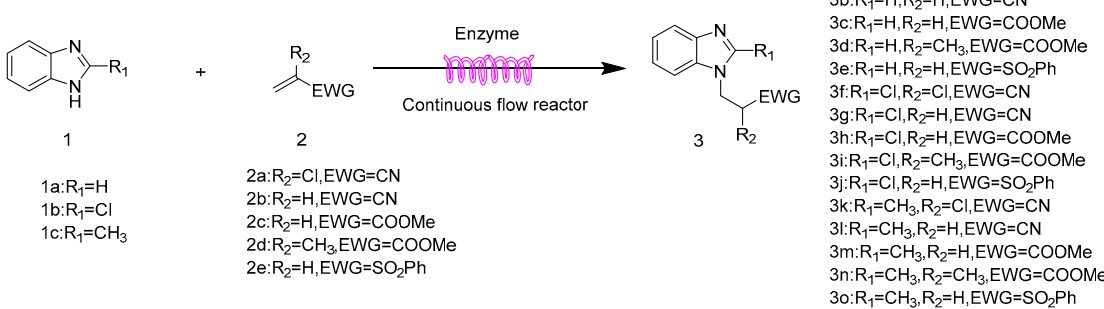

**Scheme 1.** Synthesis and design of N-substituted benzimidazole derivatives catalyzed by lipase TL IM from *Thermomyces lanuginosus* in continuous-flow microreactors.

## 2. Results

### 2.1. Effect of Reaction Solvent

As non-aqueous reaction mediums, organic solvents play a crucial role in enzyme-catalyzed synthesis reactions, which may affect the reaction results. We studied the effect of solvent on the enzymatic synthesis of N-substituted benzimidazole derivatives in continuous-flow microreactors. Benzimidazole (Table 1, 1a) reacted with 2-chloroacrylonitrile (Table 1, 2a) catalyzed by lipase TL IM from *Thermomyces lanuginosus* in continuous-flow microreactors was selected as the model reaction. Several solvents, including methanol, tert-amyl alcohol, DMSO, isopropanol, acetonitrile, n-hexane, and DMF, were investigated, and the results are shown in Table 1. As we can see from Table 1, methanol is the optimal reaction solvent for the Michael addition reaction of benzimidazole and 2-chloroacrylonitrile (Table 1, Entry 2). Therefore, methanol was selected as the reaction solvent for further research on the enzymatic synthesis of N-substituted benzimidazole derivatives in continuous-flow microreactors.

**Table 1.** The effect of solvent on the enzymatic synthesis of N-substituted benzimidazole derivatives in continuous-flow microreactors [a].

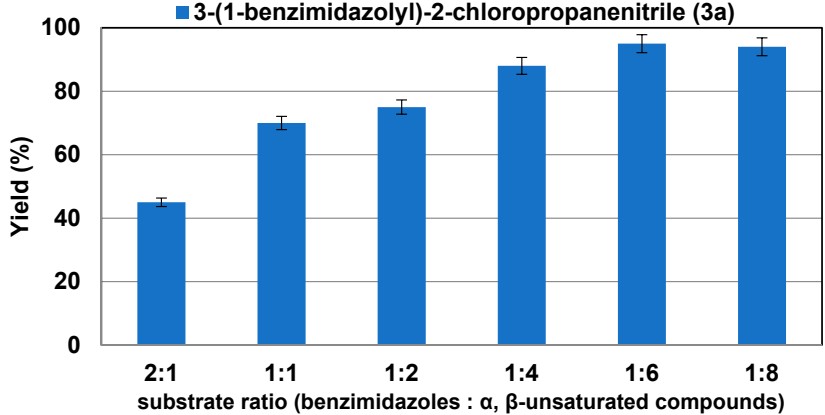

| Entry | Solvent | Catalysts | Log P | Yield [b] (%) |
|-------|---------|-----------|-------|-----------|
| 1 | Methanol | None | −0.76 | n.d. |
| 2 | Methanol | Lipozyme TL IM | −0.76 | 95.4 ± 1.6 |
| 3 | Tert-amyl alcohol | Lipozyme TL IM | 1.04 | 45.8 ± 1.5 |
| 4 | DMSO | Lipozyme TL IM | −1.3 | 68.2 ± 1.0 |
| 5 | Isopropanol | Lipozyme TL IM | 0.39 | 32.3 ± 0.5 |
| 6 | Acetonitrile | Lipozyme TL IM | −0.33 | 42.5 ± 0.8 |
| 7 | n-Hexane | Lipozyme TL IM | 3.94 | 66.4 ± 1.1 |
| 8 | DMF | Lipozyme TL IM | −1.0 | 43.2 ± 0.7 |

[a] General experimental conditions: in the continuous flow reactors, feed 1, 10 mL solvent contained 5.0 mmol benzimidazole (1a); feed 2, 10 mL solvent contained 30.0 mmol 2-chloroacrylonitrile (2a), 45 °C, flow rate 17.8 μL min$^{-1}$, residence time 35 min, enzyme 870 mg. [b] Isolated yield. Yield: 100 × (actual received amount/ideal calculated amount). The data are presented as average ± SD of triplicate experiments. n.d. means no reaction was found.

## 2.2. Effect of Substrate Ratio

As for enzyme-catalyzed reactions, the molar ratio of the reactants affects the catalytic efficiency of the enzyme and the final product yields. In this study, substrate molar ratios (benzimidazoles: α, β-unsaturated compounds) from 2:1 to 1:8 were studied, the results are shown in Figure 2. From Figure 2, we can find that the yield of the target product increases gradually with the increase in α, β-unsaturated compounds. The highest yield 95% is obtained when the substrate molar ratio (benzimidazoles: α, β-unsaturated compounds) is 1:6. At this time, continuing to increase the substrate molar ratio (benzimidazoles: α, β-unsaturated compounds) leads to the production of by-product and affects the yields. Therefore, we chose benzimidazoles: α, β-unsaturated compounds = 1:6 as our optimal substrate molar ratio in the further parameter research.

**Figure 2.** The influence of substrate ratio (benzimidazoles: α, β-unsaturated compounds) on the synthesis of N-substituted benzimidazole derivatives catalyzed by lipase TL IM from *Thermomyces lanuginosus* in continuous-flow microreactors.

### 2.3. Effect of Reaction Temperature

The rate of chemical reactions increases with temperature, but enzymes are proteins and temperature affects the responses of enzymes mainly by affecting their activity. When the temperature is low, the reaction rate increases with increasing temperature. However, when the temperature exceeds a specific range, the thermal denaturation factor of the enzyme dominates, and the reaction speed slows down with the increase in temperature. In methanol solvent, we adjusted the reaction temperature from 35 °C to 55 °C, and the results at different temperatures with 35 min residence time are shown in Figure 3. The best yield of 95% is obtained when the reaction temperature is 45 °C, and the yield decreases when the temperature continues to increase. Therefore, we chose 45 °C as the optimal reaction temperature.

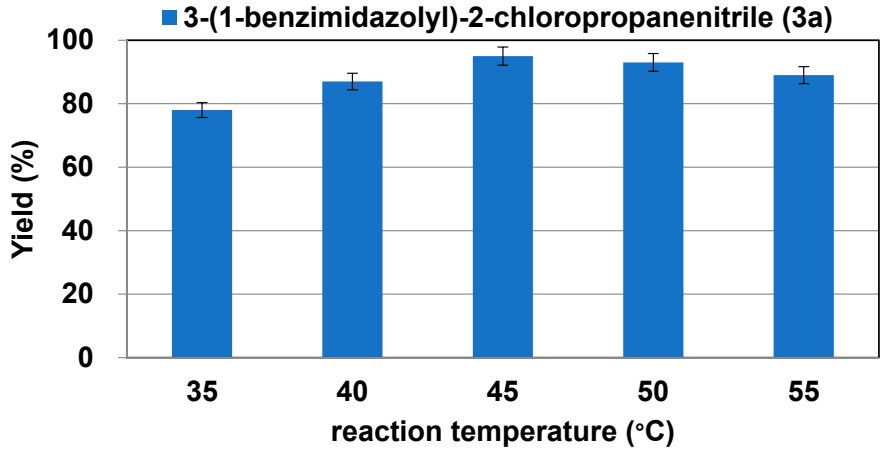

**Figure 3.** The influence of reaction temperature on the synthesis of N-substituted benzimidazole derivatives catalyzed by lipase TL IM from *Thermomyces lanuginosus* in continuous-flow microreactors.

### 2.4. Effect of Residence Time

A crucial variable in the microfluidic process is residence time. Insufficient reaction time affects the contact reaction between reactants and enzymes. If the reaction time is too long, side reactions may occur, and undesirable by-products may be generated. The effect of residence time on the lipase-catalyzed synthesis of N-substituted benzimidazole derivatives was studied by increasing the residence time from 20 to 50 min, and the results are shown in Figure 4. As we can see from Figure 4, the best yield is reached in 35 min; continuing to prolong the reaction time does not improve the reaction yield much. Therefore, 35 min was selected as the optimal residence time in the following study.

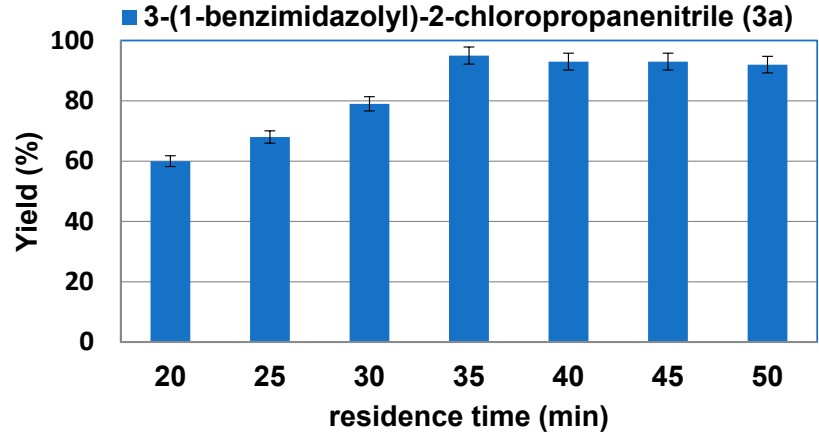

**Figure 4.** The influence of residence time on the synthesis of N-substituted benzimidazole derivatives catalyzed by lipase TL IM from Thermomyces lanuginosus in continuous-flow microreactors.

### 2.5. The Effect of Enzyme Reusability

Since immobilized lipase can further reduce costs by allowing for recycling and reuse, we explored the reproducible amount of lipase TL IM under optimal conditions. The enzyme was reused ten times, with yields of N-substituted benzimidazole derivatives above 57% (Figure 5). The short reaction time, coupled with the excellent reuse potential of the lipozyme, may lead to significant productivity gains in the synthesis of related benzimidazole derivatives.

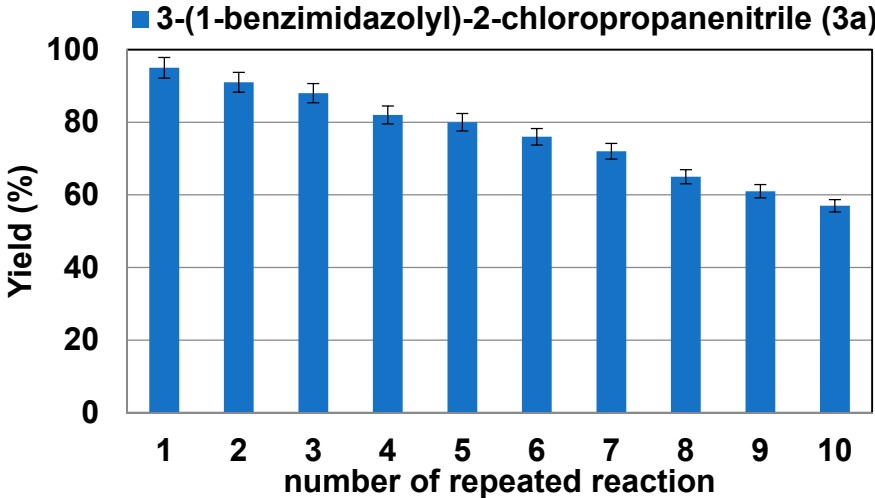

**Figure 5.** The influence of number of repeated reaction on the synthesis of N-substituted benzimidazole derivatives catalyzed by lipase TL IM from Thermomyces lanuginosus in continuous-flow microreactors.

### 2.6. The Scope and Limitation of the Synthesis of N-Substituted Benzimidazole Derivatives Catalyzed by Lipozyme TL IM in Continuous-Flow Microreactors

Finally, to explore the scope and limitations of this new high-speed aza-Michael addition of benzimidazole derivatives to $\alpha$, $\beta$-unsaturated compounds, 3 benzimidazoles (benzimidazole, 2-chlorobenzimidazole, 2-methylbenzimidazole) and 5 $\alpha$, $\beta$-unsaturated compounds (2-chloroacrylonitrile, acrylonitrile, methyl acrylate, methyl methacrylate, phenyl vinyl sulfone) were subjected to the general reaction conditions, using continuous-flow microreactor processing. The corresponding adducts were synthesized parallelly in a single experiment using lipase-catalyzed aza-Michael addition of benzimidazoles to $\alpha$, $\beta$-unsaturated compounds under continuous flow conditions, which proves that good scalability of this process (Table 2). We find that the yields decrease when there are electron-withdrawing groups on the benzimidazole ring, because the nucleophilicity of the nitrogen atom can be weakened (Table 2, Entry 6–8 and 10), while the yields are basically not affected when there are electron-donating groups on the benzimidazole ring (Table 2, Entry 11–13 and 15). We also study the effects of different $\alpha$, $\beta$-unsaturated compounds on the reaction and find that phenyl vinyl sulfone is the best reactant, as the phenyl vinyl sulfone derivatives have the best yields. The reaction yield of 2-chloroacrylonitrile is higher than acrylonitrile, which may be due to the enhancement of electropositivity of $\beta$-carbon atom by electron-withdrawing groups. Due to the steric hindrance effect, the reaction yield of methyl methacrylate is trace.

**Table 2.** Continuous process for the synthesis of N-substituted benzimidazole derivatives catalyzed by lipase TL IM from *Thermomyces lanuginosus* in continuous-flow microreactors [a].

| Entry | $R_1$ | $R_2$ | EWG | Product | Yield [b] (%) |
|-------|-------|-------|-----|---------|-----------|
| 1 | H | Cl | CN | **3a** | $95.4 \pm 1.6$ |
| 2 | H | H | CN | **3b** | $90.3 \pm 0.8$ |
| 3 | H | H | COOMe | **3c** | $92.2 \pm 1.5$ |
| 4 | H | $CH_3$ | COOMe | **3d** | trace |
| 5 | H | H | $SO_2Ph$ | **3e** | $97.1 \pm 1.2$ |
| 6 | Cl | Cl | CN | **3f** | $80.5 \pm 0.5$ |
| 7 | Cl | H | CN | **3g** | $76.8 \pm 0.9$ |
| 8 | Cl | H | COOMe | **3h** | $77.4 \pm 1.1$ |
| 9 | Cl | $CH_3$ | COOMe | **3i** | trace |
| 10 | Cl | H | $SO_2Ph$ | **3j** | $80.3 \pm 0.7$ |
| 11 | $CH_3$ | Cl | CN | **3k** | $95.6 \pm 1.4$ |
| 12 | $CH_3$ | H | CN | **3l** | $90.4 \pm 1.1$ |
| 13 | $CH_3$ | H | COOMe | **3m** | $93.2 \pm 0.6$ |
| 14 | $CH_3$ | $CH_3$ | COOMe | **3n** | trace |
| 15 | $CH_3$ | H | $SO_2Ph$ | **3o** | $97.2 \pm 0.8$ |

[a] General experimental conditions: in the continuous flow reactors, feed 1, 10 mL solvent contained 5.0 mmol benzimidazoles (**1**); feed 2, 10 mL solvent contained 30.0 mmol α, β-unsaturated compounds (**2**), 45 °C, flow rate 17.8 μL min$^{-1}$, residence time 35 min, enzyme 870 mg. [b] Isolated yield. Yield: $100 \times$ (actual received amount/ideal calculated amount). The data are presented as average $\pm$ SD of triplicate experiments.

## 3. Materials and Methods

### 3.1. Materials

All chemicals in this study were obtained from commercial sources and did not require further purification. Lipozyme TL IM from *Thermomyces lanuginosus* was purchased from Novo Nordisk (Copenhagen, Denmark). Benzimidazole, 2-chlorobenzimidazole, 2-methylbenzimidazole, 2-chloroacrylonitrile, acrylonitrile, methyl acrylate, methyl methacrylate, phenyl vinyl sulfone were purchased from Aladdin (Shanghai, China). Harvard Instrument PHD 2000 syringe pump was purchased from Harvard University (Holliston, MA, USA). The flow reactor and Y-mixer were purchased from Beijing Haigui Medical Engineering Design Co., Ltd. (Beijing China). A 400 MHz NMR spectrometer (Billerica, MA, USA) were also used in this study.

### 3.2. Experimental Setup and Experiment Conditions

A continuous-flow protocol for the aza-Michael addition of benzimidazoles to α, β-unsaturated compounds in microreactors is described in Figure 6 as well as in Figure S1. The experimental setup consists of two syringe pumps, coil reactor 1, coil reactor 2, and Y-shaped mixers (φ = 1.8 mm). Syringe pumps (Harvard apparatus PHD 2000) were used to introduce separate feed streams to 3.1 mL PFA coil reactors (2.0 mm I.D.). Silica gel tubes were filled with lipase TL IM and immersed in a constant temperature water bath to control the temperature. A total of 5 mmol of benzimidazole derivative was dissolved in 10 mL of methanol (feed 1), and 30 mmol of α, β-unsaturated compounds was dissolved in 10 mL of methanol (feed 2). Feeds 1 and 2 were placed in separate 10 mL feeders and mixed at a flow rate of 8.91 μL min$^{-1}$ in a Y-mixer at 45 °C. The resulting stream (17.8 μL min$^{-1}$) was connected to a sample vial for collection of the final mixture. The final mixture was then evaporated and the residue was separated by silica gel chromatography (200–300 mesh). Grades containing the major product were combined and the solvent was evaporated. The main products were determined by $^1$H NMR and $^{13}$C NMR.

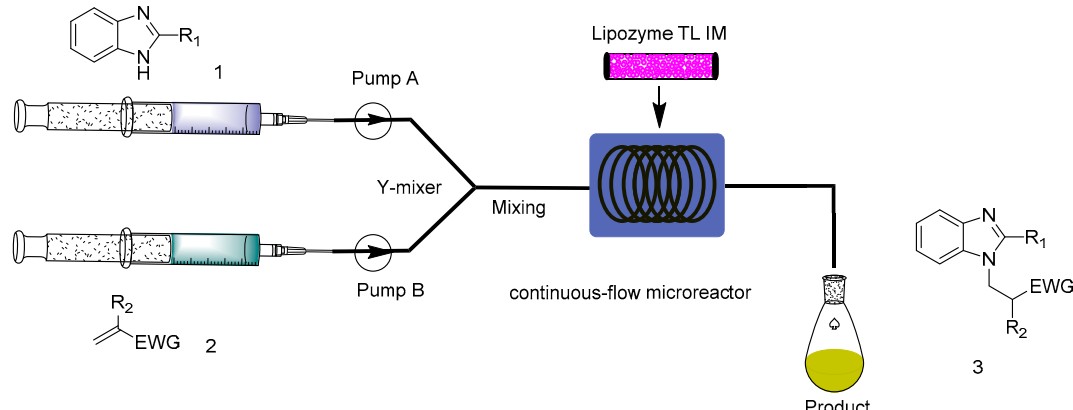

**Figure 6.** Equipment for the synthesis of N-substituted benzimidazole derivatives from benzimidazoles and α, β-unsaturated compounds catalyzed by Lipozyme TL IM from *Thermomyces lanuginosus* under continuous-flow microreactors.

### 3.3. Analytical Methods

#### 3.3.1. Thin-Layer Chromatography (TLC)

TLC analysis was performed with methanol/dichloromethane 1/20 (*v*/*v*) as the eluent. The results were detected by UV irradiation at 254 nm.

#### 3.3.2. Nuclear Magnetic Resonance (NMR) and High-Resolution Mass Spectrometry (HRMS)

The product obtained by column chromatography separation and purification was subjected to $^1$H NMR, $^{13}$C NMR, and HRMS structure confirmation.

*3-(1H-benzo[d]imidazol-1-yl)-2-chloropropanenitrile* (**3a**). White solid. $^1$H NMR (400 MHz, DMSO-*d6*) δ 8.30 (s, 1H), 7.80 (dt, *J* = 8.1, 1.0 Hz, 1H), 7.72–7.65 (m, 1H), 7.27 (dddd, *J* = 21.6, 8.4, 7.2, 1.2 Hz, 2H), 5.77 (t, *J* = 6.2 Hz, 1H), 5.04 (dd, *J* = 6.2, 1.4 Hz, 2H). $^{13}$C NMR (101 MHz, DMSO-*d6*) δ 144.64, 143.13, 133.75, 122.93, 122.29, 119.63, 116.64, 111.17, 46.91, 42.68. HRMS (ESI): calculated for $C_{10}H_9ClN_3$ [M + H]$^+$: 206.0480, found 206.0474.

*3-(1H-benzo[d]imidazol-1-yl)propanenitrile* (**3b**). White solid. $^1$H NMR (400 MHz, DMSO-*d6*) δ 8.28 (s, 1H), 7.69 (ddt, *J* = 16.9, 7.8, 0.9 Hz, 2H), 7.26 (dddd, *J* = 25.2, 8.3, 7.2, 1.2 Hz, 2H), 4.58 (t, *J* = 6.5 Hz, 2H), 3.12 (t, *J* = 6.5 Hz, 2H). $^{13}$C NMR (101 MHz, DMSO-*d6*) δ 144.06, 143.38, 133.44, 122.68, 121.97, 119.59, 118.64, 110.62, 39.91, 18.56. HRMS (ESI): calculated for $C_{10}H_{10}N_3$ [M + H]$^+$: 172.0869, found 172.0863.

*methyl 3-(1H-benzo[d]imidazol-1-yl)propanoate* (**3c**). Transparent liquid. $^1$H NMR (400 MHz, DMSO-*d6*) δ 8.21 (s, 1H), 7.69–7.59 (m, 2H), 7.23 (ddd, *J* = 14.6, 7.8, 1.3 Hz, 2H), 4.50 (t, *J* = 6.7 Hz, 2H), 3.56 (s, 3H), 2.93 (t, *J* = 6.7 Hz, 2H). $^{13}$C NMR (101 MHz, DMSO-*d6*) δ 171.28, 144.21, 143.42, 133.57, 122.39, 121.59, 119.50, 110.43, 51.61, 39.99, 33.83. HRMS (ESI): calculated for $C_{11}H_{12}N_2O_2$ [M + H]$^+$: 205.0982, found 205.0974.

*1-(2-(phenylsulfonyl)ethyl)-1H-benzo[d]imidazole* (**3e**). White solid. $^1$H NMR (400 MHz, DMSO-*d6*) δ 8.13 (s, 1H), 7.85–7.79 (m, 2H), 7.70–7.63 (m, 1H), 7.60–7.51 (m, 3H), 7.45 (dt, *J* = 8.2, 0.9 Hz, 1H), 7.26–7.14 (m, 2H), 4.59 (t, *J* = 6.6 Hz, 2H), 4.00 (t, *J* = 6.6 Hz, 2H). $^{13}$C NMR (101 MHz, DMSO-*d6*) δ 144.12, 143.29, 138.76, 133.96, 133.27, 129.36, 127.41, 122.40, 121.66, 119.44, 110.27, 53.54, 38.19. HRMS (ESI): calculated for $C_{15}H_{15}N_2O_2S$ [M + H]$^+$: 287.0849, found 287.0836.

*2-chloro-3-(2-chloro-1H-benzo[d]imidazol-1-yl)propanenitrile* (**3f**). Transparent liquid. $^1$H NMR (400 MHz, DMSO-*d6*) δ 7.86–7.80 (m, 1H), 7.64 (dd, *J* = 7.5, 1.1 Hz, 1H), 7.32 (dtd, *J* = 21.4, 7.5, 1.3 Hz, 2H), 5.78 (dd, *J* = 7.3, 6.0 Hz, 1H), 5.03 (qd, *J* = 15.5, 6.6 Hz, 2H). $^{13}$C NMR (101 MHz, DMSO-*d6*) δ 140.98, 140.42, 134.81, 123.39, 123.09, 118.75, 116.28, 111.40, 46.47, 41.66. HRMS (ESI): calculated for $C_{10}H_8Cl_2N_3$ [M + H]$^+$: 240.0090, found 240.0088.

*3-(2-chloro-1H-benzo[d]imidazol-1-yl)propanenitrile* (**3g**). White solid. $^1$H NMR (400 MHz, DMSO-*d6*) δ 7.75–7.68 (m, 1H), 7.66–7.59 (m, 1H), 7.31 (dtd, *J* = 21.2, 7.4, 1.3 Hz, 2H), 4.59 (t, *J* = 6.4 Hz, 2H), 3.10 (t, *J* = 6.4 Hz, 2H). $^{13}$C NMR (101 MHz, DMSO-*d6*) δ 141.19, 140.03,

134.69, 123.37, 122.94, 118.84, 118.36, 110.94, 48.73, 17.75. HRMS (ESI): calculated for $C_{10}H_9ClN_3$ [M + H]$^+$: 206.0480, found 206.0474.

*methyl 3-(2-chloro-1H-benzo[d]imidazol-1-yl)propanoate* (**3h**). Transparent liquid. $^1$H NMR (400 MHz, DMSO-$d_6$) δ 7.72–7.54 (m, 2H), 7.27 (dtd, *J* = 23.4, 7.3, 1.3 Hz, 2H), 4.51 (t, *J* = 6.9 Hz, 2H), 3.37 (d, *J* = 1.4 Hz, 3H), 2.87 (t, *J* = 6.9 Hz, 2H). $^{13}$C NMR (101 MHz, DMSO-$d_6$) δ 170.79, 141.13, 139.90, 134.73, 123.03, 122.49, 118.65, 110.75, 51.66, 39.95, 33.13. HRMS (ESI): calculated for $C_{11}H_{12}ClN_2O_2$ [M + H]$^+$: 239.0582, found 239.0577.

*2-chloro-1-(2-(phenylsulfonyl)ethyl)-1H-benzo[d]imidazole* (**3j**). White solid. $^1$H NMR (400 MHz, DMSO-$d_6$) δ 7.86–7.79 (m, 2H), 7.72–7.63 (m, 1H), 7.59–7.46 (m, 4H), 7.25 (dtd, *J* = 19.2, 7.4, 1.3 Hz, 2H), 4.58 (t, *J* = 6.7 Hz, 2H), 3.99 (t, *J* = 6.7 Hz, 2H). $^{13}$C NMR (101 MHz, DMSO-$d_6$) δ 141.01, 139.73, 138.59, 134.44, 134.01, 129.35, 127.30, 123.01, 122.58, 118.59, 110.71, 52.49, 38.02. HRMS (ESI): calculated for $C_{15}H_{14}ClN_2O_2S$ [M + H]$^+$: 321.0459, found 321.0456.

*2-chloro-3-(2-methyl-1H-benzo[d]imidazol-1-yl)propanenitrile* (**3k**). White solid. $^1$H NMR (400 MHz, DMSO-$d_6$) δ 7.75–7.65 (m, 1H), 7.59–7.48 (m, 1H), 7.25–7.13 (m, 2H), 5.78–5.70 (m, 1H), 5.02 (dd, *J* = 15.4, 6.2 Hz, 1H), 4.93 (dd, *J* = 15.4, 7.7 Hz, 1H), 2.61 (s, 3H). $^{13}$C NMR (101 MHz, DMSO-$d_6$) δ 152.29, 142.20, 134.95, 121.88, 121.85, 118.30, 116.69, 110.70, 45.81, 42.01, 13.77. HRMS (ESI): calculated for $C_{11}H_{11}ClN_3$ [M + H]$^+$: 220.0636, found 220.0634.

*3-(2-methyl-1H-benzo[d]imidazol-1-yl)propanenitrile* (**3l**). White solid. $^1$H NMR (400 MHz, DMSO-$d_6$) δ 7.65–7.58 (m, 1H), 7.58–7.49 (m, 1H), 7.18 (pd, *J* = 7.2, 1.4 Hz, 2H), 4.52 (t, *J* = 6.6 Hz, 2H), 3.05 (t, *J* = 6.6 Hz, 2H), 2.59 (s, 3H). $^{13}$C NMR (101 MHz, DMSO-$d_6$) δ 151.84, 142.36, 134.70, 121.69, 121.57, 118.80, 118.31, 110.04, 38.72, 17.87, 13.50. HRMS (ESI): calculated for $C_{11}H_{12}N_3$ [M + H]$^+$: 186.1026, found 186.1020.

*methyl 3-(2-methyl-1H-benzo[d]imidazol-1-yl)propanoate* (**3m**). Transparent liquid. $^1$H NMR (400 MHz, DMSO-$d_6$) δ 7.58–7.45 (m, 2H), 7.16 (pd, *J* = 7.2, 1.4 Hz, 2H), 4.43 (t, *J* = 7.0 Hz, 2H), 3.56 (s, 3H), 2.85 (t, *J* = 6.9 Hz, 2H), 2.55 (s, 3H). $^{13}$C NMR (101 MHz, DMSO-$d_6$) δ 171.24, 151.81, 142.43, 134.75, 121.48, 121.25, 118.24, 109.86, 51.56, 38.99, 33.35, 13.40. HRMS (ESI): calculated for $C_{12}H_{14}N_2O_2$ [M + H]$^+$: 219.1129, found 219.1118.

*2-methyl-1-(2-(phenylsulfonyl)ethyl)-1H-benzo[d]imidazole* (**3o**). White solid. $^1$H NMR (400 MHz, DMSO-$d_6$) δ 7.90–7.82 (m, 2H), 7.74–7.65 (m, 1H), 7.57 (t, *J* = 7.8 Hz, 2H), 7.48–7.40 (m, 1H), 7.33–7.25 (m, 1H), 7.17–7.06 (m, 2H), 4.49 (t, *J* = 6.8 Hz, 2H), 3.93 (t, *J* = 6.8 Hz, 2H), 2.46 (s, 3H). $^{13}$C NMR (101 MHz, DMSO-$d_6$) δ 151.68, 142.28, 138.76, 134.49, 133.99, 129.38, 127.35, 121.53, 121.38, 118.20, 109.68, 52.95, 36.97, 13.36. HRMS (EI): calculated for $C_{16}H_{16}N_2O_2S$ [M]$^+$: 300.0932, found 300.0903.

## 4. Conclusions

In summary, a microfluidic biocatalysis system applied for the synthesis of N-substituted benzimidazole derivatives by aza-Michael addition was developed with sustainable and convenient manipulation. Benzimidazoles (benzimidazole, 2-chlorobenzimidazole, 2-methylbenzimidazole), as less nucleophilic agents, reacted with α, β-unsaturated compounds (acrylonitriles, acrylate esters, phenyl vinyl sulfone) catalyzed by lipase TL IM from *Thermomyces lanuginosus* in continuous-flow microreactors were studied. Reaction parameters including solvent, substrate ratio, reaction temperature, reactant donor/acceptor structures, and reaction time were investigated. In this continuous-flow enzymatic strategy, a series of N-substituted benzimidazole derivatives are synthesized and characterized, which proves that the microfluidic enzymatic process in aza-Michael addition has good scalability. The salient feature of this method, including mild reaction conditions (methanol, 45 °C), short reaction time (35 min), easily available, and reusable catalyst, made this approach a promising fast synthesis strategy for further research to develop novel and highly potent active drugs. In our further research, we plan to develop a two-step tandem synthesis method of benzimidazole propionamide derivatives. Benzimidazole propionamide derivatives were reported as having anti-proliferative activity against human prostate cancer and potential peptidyl-prolyl *cis/trans* isomerase Pin1 inhibitory activity, and could be introduced into drug molecules to prepare novel anti-tumor drugs. Synthesis of N-

substituted benzimidazole derivatives by aza-Michael addition developed in this work laid a preliminary foundation for further research on drugs of benzimidazole propionamide.

**Supplementary Materials:** The following supporting information can be downloaded at: https://www.mdpi.com/article/10.3390/catal12121658/s1, Figure S1: Continuous flow microreactor. References [22,25,49,50] are cited in the Supplementary Materials.

**Author Contributions:** R.-K.J., Y.P. and L.-H.D.: subject selection, experimental design, drafted and revised the manuscript; R.-K.J., Y.P. and L.-Y.Z.: background research and experimental optimization; Z.-K.S., S.-Y.Z., H.L., A.-Y.Z. and H.-J.X.: collected data; L.-Y.Z., L.-H.D., Z.-K.Y. and X.-P.L.: analyzed the data and revised the manuscript. All authors have read and agreed to the published version of the manuscript.

**Funding:** This research was funded by the Natural Science Foundation of Zhejiang Province and the Key Research and Development Projects of Zhejiang Province (LGN20C200020 and 2020C03090 and KYYHX-20211096), the Zhejiang Provincial Key Discipline of Chemistry Biology, the National Science and Technology Support Project (2015BAD14B0305), the National Natural Science Foundation of China (21306172), and the Science and Technology Research Program of Zhejiang Province (2014C32094), and the Natural Science Foundation of Zhejiang University of Technology (116004029).

**Data Availability Statement:** Not applicable.

**Acknowledgments:** We thank the Natural Science Foundation of Zhejiang Province and the Key Research and Development Projects of Zhejiang Province (LGN20C200020 and 2020C03090 and KYYHX-20211096), the Zhejiang Provincial Key Discipline of Chemistry Biology, the National Science and Technology Support Project (2015BAD14B0305), the National Natural Science Foundation of China (21306172), and the Science and Technology Research Program of Zhejiang Province (2014C32094), as well as the Natural Science Foundation of Zhejiang University of Technology (116004029) for financial support.

**Conflicts of Interest:** The authors declare no conflict of interest.

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
