# Peer review of "Microfluidics Biocatalysis System Applied for the Synthesis of N-Substituted Benzimidazole Derivatives by Aza-Michael Addition"

_catalysts, doi:10.3390/catal12121658_

Round 1
Reviewer 1 Report
The submission by Pan, et al presents a microfluidic system that incorporates a commercially available, immobilized Lipase for the synthesis of benzimidazole derivatives.
The authors demonstrate reasonable substrate scope of the catalyst and also investigate various parameters (temperature, residence time, etc.) and reuse (up to 10X) of the microfluidic system. This paper is well written and, in principle, comprises a nice addition to the biocatalysis literature.
However, prior to publication, there are three major concerns that the authors should address:
1. The presentation of the microfluidic system is inadequate. A simple cartoon is not sufficient for reproducibility. At a minimum, detailed photographs (not cartoons) and specifications/measurements should be provided in the supporting information.
2. There is very nice NMR data provided for each product. However, no mass spectrometry data is available. This should be provided for each product.
3. In some products, a new chiral center is produced (ex. 3k, 3l). Is any enantioselectivity observed or are the products racemic?
Reviewer 2 Report
The manuscript entitled "Microfluidics Biocatalysis System Applied for the Synthesis of N-substituted Benzimidazole Derivatives by aza-Michael Addition " was submitted for its publication in Catalyst. In this work, the authors describe the use of the lipase TL IM from Thermomyces lanuginosus in continuous-flow microreactors to prepare a large number of compounds, although it is not specified if they are unreported compounds.
However, the subject that is addressed is a topic that has been widely studied and reported in the bibliography, and the only novelty is the use of benzimidazole, 2-methylbenzimidazole and 2-chlorobenzimidazole as substrates. The research group has already similar reports from other substrates and in particular with 6-nitro benzimidazole (Org. Biomol. Chem., 2019, 17, 807-812). There is no originality neither in the biocatalyst nor in the methodology nor in the experimental procedure. In addition, the text would require a thorough revision of the language. Therefore, from my point of view, I do not recommend its publication.
Special comments
1. “α, β-unsaturated alkenes” is not a correct expression. It must be replace by α,β-unsaturated compounds or conjugated alkenes
2. “All in all, enzyme-catalyzed specificity combined with continuous flow reaction technology was first applied to the synthesis of N-substituted benzimidazole derivatives...” in the abstract and in a similar sentence in the conclusions section is not true since the authors previously reported the synthesis of other N-substituted benzimidazole derivatives
3. Since there are many citations in the literature on the use of the lipase TL IM from Thermomyces lanuginosus as biocatalysts in Michael addition, they should be made explicit in the introduction.
4. What can be said about the stereoselectivity of the stereocenter formed in the products?
5. The conclusions drawn from Table 2 should be reviewed. There are no significant differences in yields between benzimidazole and 2-methylbenzimidazole and no comment is made on the phenylvinylsulfone derivatives which appear to have the best yields.
Reviewer 3 Report
Manuscript by Luo et al. reports the synthesis of N-substituted benzimidazole derivatives from benzimidazoles and α, β-unsaturated alkenes by using Lipozyme TL IM in continuous-flow microreactors. The protocol is synthetically attractive in accessing N-substituted benzimidazole. The substrate scope is demonstrated by using different α, β-unsaturated alkenes with high yields. I would suggest the publications after authors address the following issues.
1. The authors are encouraged to probe whether Methyl vinyl ketone, Methyl vinyl sulfone and Acrylophenone could be used in the reactions.
2. The scalability of this process should be commented on.
3. In table 2, entries 4, 9 and 14 are same (In all entries R1 is H). Please correct these.
4. In supporting, same correction need to be require in table 1.
Round 2
Reviewer 1 Report
The authors have made the requested additions to the manuscript and have significantly improved its overall quality. I am willing to accept the article in its present form.
Reviewer 2 Report
The manuscript entitled "Microfluidics Biocatalysis System Applied for the Synthesis of N-substituted Benzimidazole Derivatives by aza-Michael Addition " has been modified taking into account the suggestions made by the reviewers. This version has been improved enough to be published in Catalysts.
Minor comment
Line 217 pag. 7 “ And the reaction yield of 2-acrylonitrile is higher than acrylonitrile” must be changed by “And the reaction yield of 2-chloroacrylonitrile is higher than acrylonitrile”
Reviewer 3 Report
Manuscript has been sufficiently improved to warrant publication in Catalysts. I would recommend the manuscript for publication in Catalysts .